# TOF-Based Fast Self-Positioning Algorithm for UWB Mobile Base Stations

**DOI:** 10.3390/s21196359

**Published:** 2021-09-23

**Authors:** Yuxiang Han, Xiaoming Zhang, Zhengxi Lai, Yuchen Geng

**Affiliations:** 1National Key Laboratory for Electronic Measurement Technology, North University of China, Taiyuan 030051, China; s1906126@st.nuc.edu.cn (Y.H.); s1906121@st.nuc.edu.cn (Y.G.); 2Key Laboratory of Instrumentation Science & Dynamic Measure, Ministry of Education, North University of China, Taiyuan 030051, China; 3Shenzhen ImyFit Technology Co., Ltd., Shenzhen 518107, China; laizx@cositea.com

**Keywords:** Ultra-Wideband (UWB), Time of Flight (TOF), fast self-positioning, base station layout

## Abstract

To solve the problem of heavy workload and high cost when acquiring the position of Ultra-Wideband (UWB) mobile base stations in sports fields, a fast self-positioning algorithm for UWB mobile base stations algorithm based on Time of Flight (TOF) is proposed. First, according to the layout of the base stations in the sports field, the local coordinate system is determined, and an equation based on the ranging information between the base stations is established; the Least Square method is used to calculate the coordinates of each base station, and the Newton Iteration method is used to converge the positioning results. Then the origin and propagation law of positioning error, as well as the method of reducing the positioning error are analyzed. The simulation data and experimental results show that the average positioning accuracy of the mobile base station is within 0.05 m, which meets the expected accuracy of the base station position measurement. Compared with traditional manual measurement methods, base station self-positioning can effectively save deployment time and reduce workload.

## 1. Introduction

Emerging industries such as “smart sports” and “technological sports” have gradually become an significant part of the sports and health industry. “High-tech + sports” as an emerging cross-research discipline applies the concept of technology to sports, making the development of fitness and competitive sports present a trend of integration and digitalization of high-tech technologies.

Due to the implementation of scientific sports, the requirements for real-time and precise positioning of athletes are getting higher and higher. The positioning equipment for athletes requires accuracy of decimeter level and a data update rate greater than 10 Hz, which can accurately display the position of athletes in real-time. UWB position is carrier-free communication technology, and its data is transmitted on a nanosecond baseband narrow pulse [1]. Therefore, the system complexity and power consumption of the transceiver can be reduced. The ultra narrow pulse enables the UWB signal to have a high time resolution, which can achieve centimeter-level positioning accuracy [2]. Compared with Bluetooth [3], Wifi (Wireless Fidelity) [4], Ultrasonic [5], RFID (Radio Frequency Identification) [6], GNSS (Global Navigation Satellite System) [7,8], and other positioning technologies, UWB technology has been widely used in the field of indoor and outdoor positioning by its advantages such as low consumption, low system complexity, high multipath resolution, and high system security [9].

The accuracy of UWB base station coordinates directly affects the positioning accuracy of athletes. Therefore, accurately obtaining the position of the UWB base stations is of great significance to the positional accuracy of the athletes. However, the traditional method for acquiring the position of the base station requires manual measurement with the help of tools, and it is not convenient to manually measure the position of the base station in a larger sports field. In addition, the ideal real-time positioning system should achieve the characteristics of measuring according to the site conditions, but the traditional base station positioning method does not meet the needs for rapid deployment.

In order to solve this problem, Yang Xiaofei [10] and others introduce an automatic configuration method for UWB indoor positioning base stations. A virtual triangle is constructed by the distance relationship between a base station at an unknown location and two base stations at a known location. The relative coordinates of the unknown base station are obtained using the law of cosines. Zhang Meiyan [11] and others propose an iterative self-positioning and calibration method for UWB base stations. The *n* self-positioning base stations are sorted according to the *x*-axis coordinates in the coordinate system from small to large; the distance between the *n* self-positioning base stations and the two base stations with known positions is calculated by the Least Square method to obtain the position coordinates. Yu et al. [12] describe a self-calibration method for base station’s position for indoor positioning. The Markov State Transition Equation is used to calculate the state vector of all the coordinates between the base stations. The iterative trilateral positioning technique is used to estimate the position of the base station. By simulating the minimum ranging error, the final static error is 15 cm. The methods all simplify the base station positioning process, and the base station position self-positioning can be conveniently performed in applications that do not require high base station positioning accuracy. However it does not meet the requirements of high-precision positioning in sports fields.

In order to improve self-positioning accuracy, Vashitha [13] et al. introduce a self-calibration scheme that uses UWB pulse radio to determine the location of base stations. The positioning method uses Differential Time difference of arrival technology for ranging, and the positioning error is small within 10 m, but the system complexity is high because it uses the method of clock synchronization. M. Hamer [14] and others propose an algorithm that uses Time Difference of Arrival to estimate the position of base stations. This algorithm is strict in assumptions. It requires four base stations to be placed at known positions. In applications with more base stations, the positioning time of the base stations can be effectively reduced. However it is not suitable for personnel positioning in sports fields with a small number of base stations.

In order to quickly locate the mobile base station in the sports field, this work proposes a TOF-based [15] fast self-positioning algorithm for UWB mobile base stations. First, a local coordinate system according to the layout of base stations in the sports field is established. An equation based on the distance measurement information between base stations established. Then, the Least Square method is used to calculate the coordinates of each base station, and finally, the Newton iteration method is used to converge the positioning results. Base stations are arranged in the sports field as needed, and the relative positions of the base stations can be quickly located by communicating with each other between the base stations. This method can reduce the deployment time and cost of base stations, reduce the workload and difficulty of deploying UWB mobile base stations. The system is applied to the scene of movable base stations in sports fields, the experiment proves that the whole feasibility of the system, the accuracy of the positioning algorithm is within 0.05 m. It takes 0.2 s to complete a positioning experiment, which can quickly and accurately determine the coordinates of the movable base stations in the sports field.

## 2. Establishment of Ranging Error Model

This work uses DecaWave’s DW1000 chip for ranging experiments. Because time delay and clock drift will affect the ranging accuracy, it is necessary to establish an error model to reduce the ranging error.

### 2.1. Clock Drift

Assuming that there are *n* base stations in the sports field, any two base stations can communicate and range between them. In order to reduce the ranging error caused by the clock error of the base station, the technology of Symmetric Double-Sided Two-Way Ranging (SDS-TWR) between the base stations is carried out, as shown in Figure 1. Base station *i* initiates the first ranging message and records the sending timestamp *t*_0_; base station *j* records the receiving timestamp *t*_1_ while generating response information records the sending timestamp *t*_2_; when base station *i* receives the data, it records the timestamp *t*_3_, then base station i sends data with timestamps *t*_0_, *t*_3_, *t*_4_, base station *j* receives and records timestamp *t*_5_, and four time differences can be obtained. The base station *j* calculates the distance to the base station *i* according to the time difference [16].

The above four time differences are
(1)$$t_{\mathrm{reply}1}=t_{2}-t_{1}$$
(2)$$t_{\mathrm{reply}2}=t_{4}-t_{3}$$
(3)$$t_{\mathrm{round}1}=t_{3}-t_{0}$$
(4)$$t_{\mathrm{round}2}=t_{5}-t_{2}$$

Among that, *t*_round1_ represents the time from sending the polling signal to base station *i* to receiving the response signal of base station *j*; *t*_round2_ represents the time from sending the response signal to base station *j* to receiving the ranging information sent by base station *i*; *t*_reply1_ represents the time from base station *j* receiving the polling signal to The time when the response signal is sent; *t*_reply2_ represents the time from when the base station *i* receives the response signal to when it sends the ranging information.

Let *t* be the duration of the signal flying in the air, then
(5)$$t_{\mathrm{round}1}=2t+t_{\mathrm{reply}2}$$
(6)$$t_{\mathrm{round}2}=2t+t_{\mathrm{reply}1}$$

Then
(7)$$t=\frac{t_{\mathrm{round}1}\times t_{\mathrm{round}2}-t_{\mathrm{reply}1}\times t_{\mathrm{reply}2}}{t_{\mathrm{round}1}+t_{\mathrm{round}2}+t_{\mathrm{reply}1}+t_{\mathrm{reply}2}}$$

The distance *d* between base stations can be expressed as
(8)d=c⋅t+Δd
where *c* is the radio propagation speed and Δ*d* is the ranging error. Assuming that the clock offset errors of base station *i* and base station *j* are *e_i_* and *e_j_*, and the real-time without clock drift is *t*’’, then
(9)$$t_{\mathrm{round}1}^{’’}=(1+e_{i})t_{\mathrm{round}1}$$
(10)$$t_{\mathrm{reply}1}^{’’}=(1+e_{i})t_{\mathrm{reply}1}$$
(11)$$t_{\mathrm{round}2}^{’’}=(1+e_{j})t_{\mathrm{round}2}$$
(12)$$t_{\mathrm{reply}2}^{’’}=(1+e_{j})t_{\mathrm{reply}2}$$
(13)$$\begin{array}{l} t’’=\frac{t_{\mathrm{round}1}^{’’}\times t_{\mathrm{reply}2}^{’’}-t_{\mathrm{round}2}^{’’}\times t_{\mathrm{reply}1}^{’’}}{t_{\mathrm{round}1}^{’’}+t_{\mathrm{reply}2}^{’’}+t_{\mathrm{round}2}^{’’}+t_{\mathrm{reply}1}^{’’}}\\ =t\frac{\left(1+e_{i})+(1+e_{j}\right)}{2(1+e_{i})(1+e_{j})} \end{array}$$
(14)$$\Delta t=t-t^{’’}=\frac{e_{i}+e_{j}+2e_{i}e_{j}}{2(1+e_{i})(1+e_{j})}t$$

Assuming that the clock offset errors of the two base stations are the same because of $e_{i}\ll 1,e_{j}\ll 1$, then
(15)$$\Delta t=e_{i}t$$

If the error of the external crystal oscillator used by the chip is 20 ppm, when the transmission distance is 200 m, the propagation time of the radio in the air is about 0.6 μs, the time error caused by clock drift is
(16)$$20\times 10^{-6}\times 0.6\times 10^{-6}=12\times 10^{-12}=12\;\mathrm{ps}$$

Then the measuring distance of 200 m adopts the TOF two-way ranging principle, and the ranging error caused by clock drift is 3.6 mm.

It can be seen that the ranging error caused by clock drift is at the level of mm. Compared with TWR ranging, SDS-TWR technology has little effect on the sports field ranging result, and the impact of this error can be reduced during the calibration of the receiver delay parameters.

### 2.2. Antenna Delay Calibration

The processes of sending message frames are shown in Figure 2, the digital circuit sends a digital signal, which is modulated into an analog signal, and then the analog signal passes through the radio FC and is sent out by the antenna to receive the message frame. The process of receiving message frames is opposite to the sending process. The digital circuit will record the moment when the first bit of the PHR leaves or arrives at the digital circuit and take the time as the time stamp sent or received. The physical layer frame structure of the chip is shown in Figure 3; then, the time stamp recorded by DW1000 is not the time when the radio frequency signal leaves or arrives at the antenna, but a timestamp that includes the antenna delay [17].
(17)$$t_{\mathrm{delayT}}=t_{T}^{’}-t_{T}$$
(18)$$t_{\mathrm{delayR}}=t_{R}-t_{R}^{’}$$

Among that, *t′*_T_ represents the actual sending time of the system, *t*_T_ represents the sending timestamp of the system, *t*_R_ represents the receiving timestamp of the system, and *t′*_R_ represents the actual receiving time of the system. Assuming that the transmission delay and the reception delay are equal, that is *t*_delayT_ = *t*_delayR_ = *t*_delay_, then
(19)$$t_{R}^{’}-t_{T}^{’}=t_{R}-t_{T}-2t_{\mathrm{delay}}$$

The antenna delay parameter is obtained by experiment. Place the base station at a known location and measure the distance between base stations using SDS-TWR technology. During the measurement, the antenna delay parameter in the program is continuously adjusted. When the antenna delay parameter minimizes the error of the measured distance, the antenna delay parameter is used.

Place two base stations at two points with a known distance of 25 m, and adjust the delay parameters according to the test results. Finally, when the parameters are set to 0.3 ns, the ranging result is closest to 25 m. It can be seen that the ranging error of the base stations after the delay parameter calibration obeys a normal distribution with a mean value of 0 and a standard deviation of 0.05 m, as shown in Figure 4.

In summary, the ranging error is related to clock drift and time delay. Because the fixed location base station does not need a high data update rate, and the accuracy of SDS-TWR is high, and the ranging process is simple. Therefore, TOF based positioning method is selected.

## 3. Coordinate Calculation

The positioning method in which unknown nodes need to communicate with known nodes and other unknown nodes simultaneously is called cooperative positioning [18]. In order to verify the correctness of the observation results and improve the reliability of the results, the method of Redundant Observations in actual work is adopted. The observed results are appropriately corrected according to the principle of Least Squares, to obtain a set of the most reliable results, this process is called measurement adjustment [19]. This work uses the collaborative positioning method, and the Least Square principle of measurement adjustment is applied to base stations self-positioning. Therefore, communication between a base station with a known location (base station 1) and a base station with an unknown location is required to obtain distance information, and then determine the location of the base station; at the same time, each base station needs to perform mutual ranging to obtain ranging information. The ranging information of each pair of base stations is used for the calculation of coordinates. The advantage is that it can reduce the influence of gross errors on the measurement results. It can also minimize the error and realize the effect of reducing the influence of measurement error in the process of error transmission.

### 3.1. Coordinate System Construction

Set the two Base Stations (BS) in the sports field as BS 1 and BS 2. The line connecting BS 1 and BS 2 on the ground projection point is defined as the *x*-axis of the coordinate system, and the *y*-axis of the coordinate system is on the ground of the sports field; it passes through BS1 and is perpendicular to the *x*-axis. Assuming that the height of each base station is equal, the coordinates of BS 1 are (0, 0, *z*), the coordinates of BS 2 are (*x*_2_, 0, *z*), and the coordinates of BS *n* are (*x_n_*, *y_n_*, *z*), Then the distance between every two base stations is expressed as:(20)$$d_{ij}=\sqrt{{\left(x_{i}-x_{j}\right)}^{2}+{\left(y_{i}-y_{j}\right)}^{2}}\;i=1,2,3\ldots .n;j=1,2,3\ldots n;$$

According to the positioning accuracy requirements of the sports field and the number limit of base stations deployment, a coordinate system is established for analysis. The midpoint of the short side of the playground is the coordinate origin, and the base stations are arranged in a hexagonal shape, which is approximately a honeycomb shape, as shown in Figure 5:

According to Formula (20), the nonlinear equations about distance and base station coordinates can be obtained. In this work, the Least Square method and Newton iteration method are used to solve the final coordinates of the base stations. The specific method is as follows.

Assuming that *X_k_* is the coordinate value of the kth iteration, then
(21)$$X_{k}={\left[x_{2k},x_{3k},y_{3k}\cdots x_{nk},y_{nk}\right]}^{T}$$

Taylor expansion is used to find the Least Square solution of the equation.

Let ***f*** be the distance measurement matrix between each base station, then
(22)$$f={\left[f_{12},f_{1i}\cdots f_{ij}\right]}^{T}$$
(23)$$f_{ij}=\sqrt{{\left(x_{i}-x_{j}\right)}^{2}+{\left(y_{i}-y_{j}\right)}^{2}}$$
carrying out Taylor expansion on the nonlinear equations ***f***
(24)$$f_{ij}=f_{ij}(X_{k})+\frac{\partial f_{ij}}{\partial x_{i}}\Delta x_{i}+\frac{\partial f_{ij}}{\partial y_{i}}\Delta y_{i}+\frac{\partial f_{ij}}{\partial x_{j}}\Delta x_{j}+\frac{\partial f_{ij}}{\partial y_{j}}\Delta y_{j}$$
where
(25)$$\Delta x_{i}=x_{i}-x_{ik},\Delta y_{i}=y_{i}-y_{ik},i=2,3\cdots n$$
set the coefficient matrix *H_k_* is:(26)$$\begin{array}{l} H_{k}={\left[\begin{array}{c} \frac{\partial f_{12}}{\partial X_{k}}\\ \frac{\partial f_{i1}}{\partial X_{k}}\\ \frac{\partial f_{i2}}{\partial X_{k}}\\ \frac{\partial f_{ij}}{\partial X_{k}} \end{array}\right]}_{\frac{n(n-1)}{2}\times m} \end{array}$$
where
(27)$$\begin{array}{l} \frac{\partial f_{12}}{\partial X_{k}}={\left[{\begin{array}{cc} 1 & 0 \end{array}}_{1}{}_{\times (2n-4)}\right]}_{1\times (2n-3)}\\ \frac{\partial f_{i1}}{\partial X_{k}}={\left[\begin{array}{cccc} 0_{1}{}_{\times (2i-5)} & \frac{x_{i}}{f_{i1}} & \frac{y_{i}}{f_{i1}} & 0_{1}{}_{\times 2(n-i)} \end{array}\right]}_{\left(n-1)\times (2n-3\right)}\\ \frac{\partial f_{i2}}{\partial X_{k}}={\left[\begin{array}{ccccc} \frac{x_{2}-x_{i}}{f_{i2}} & 0_{1}{}_{\times (2i-6)} & \frac{x_{i}-x_{2}}{f_{i2}} & \frac{y_{i}}{f_{i2}} & 0_{1}{}_{\times 2(n-i)} \end{array}\right]}_{\left(n-2)\times (2n-3\right)}\\ \frac{\partial f_{ij}}{\partial X_{k}}={\left[\begin{array}{ccccccc} 0_{1}{}_{\times (2i-5)} & \begin{array}{c} \frac{x_{i}-x_{j}}{f_{ij}} \end{array} & \frac{y_{i}-y_{j}}{f_{ij}} & 0_{1}{}_{\times 2(j-i-1)} & \frac{x_{j}-x_{i}}{f_{ij}} & \frac{y_{j}-y_{i}}{f_{ij}} & 0_{1}{}_{\times 2(n-j)} \end{array}\right]}_{\left(n-2)(n-3)\times (2n-3\right)} \end{array}$$

The initial iterative value of *X_k_* is obtained by solving the nonlinear equations.
(28)$$\begin{array}{l} x_{20}=d_{12}{}^{2}\\ x_{i0}=\frac{M}{2d_{12}{}^{2}}\\ y_{i0}=\frac{\sqrt{4d_{12}{}^{2}d_{1i}{}^{2}-M^{2}}}{2d_{12}{}^{2}} \end{array}$$
where
(29)$$M=d_{1i}{}^{2}-d_{2i}{}^{2}+d_{12}{}^{2}$$
set
(30)$$\Delta X_{k}=[\Delta x_{2k},\Delta x_{3k},\Delta y_{3k}\cdots \Delta x_{ik},\Delta y_{ik}\cdots \Delta x_{nk},\Delta y_{nk}{]}^{T}$$
(31)$$\Delta f_{k}=f-f_{k}$$
(32)$$f_{k}={\left[f_{12}{|}_{X_{k}},f_{1i}{|}_{X_{k}}\cdots f_{ij}{|}_{X_{k}}\right]}^{T}$$
thus
(33)$$\Delta f_{k}=H_{k}\Delta X_{k}$$

Solve the Δ*X_k_* using the Least Square method
(34)$$\Delta X_{k}={\left(H_{k}^{T}H_{k}\right)}^{-1}H_{k}^{T}\Delta f_{k}$$
the *X*_1_ in the first iteration is
(35)$$X_{1}=\Delta X+X_{0}$$

Solve for Δ*X_k_* by the method of Newton iteration, stop the iteration when Δ*X_k_* is less than the error threshold; at this time, *X_k_*_+1_ is
(36)$$X_{k+1}=X_{k}+\Delta X_{k}$$

The final base station’s coordinates are the values corresponding to the *X_k_*_+1_ matrix.

### 3.2. Evaluating Indicator

(i)Error Coefficient

Assuming that the ranging error follows a normal distribution with the mean of μ_D_ and a standard deviation of σ*_D_*, then Δ***f****_k_* follows a normal distribution with a mean value of 0 and a standard deviation of $\sqrt{2}$σ*_D_*.

The expectation of base stations coordinate Δ*X_k_* is as follows
(37)$$E(\Delta X_{k})={\left(H_{k}^{T}H_{k}\right)}^{-1}H_{k}^{T}E(\Delta f_{k})=0$$

The variance of base stations coordinate Δ*X_k_* is as follows
(38)$$D(\Delta X_{k})=E(\Delta X_{k}\Delta X_{k}^{T})=2{\left(H_{k}^{T}H_{k}\right)}^{-1}\sigma_{D}^{2}$$

Take ($H_{k}^{T}H_{k}$)^−1^ as the coefficient matrix of the influence of TOF ranging error on the self-location accuracy of the mobile base station. The coefficient matrix can be obtained by the following formula.
(39)$${\left(H_{k}^{T}H_{k}\right)}^{-1}=\left[\begin{array}{cccc} Q_{11} & Q_{12} & \cdots & Q_{1(2n-3)}\\ Q_{21} & Q_{22} & \cdots & Q_{2(2n-3)}\\ \vdots & \vdots & \ddots & \vdots \\ Q_{(2n-3)1} & Q_{(2n-3)2} & \cdots & Q_{\left(2n-3)(2n-3\right)} \end{array}\right]$$

The error coefficient at the coordinates of each base station can be expressed as follows,
(40)$$\begin{array}{l} Px_{2}=\sqrt{Q_{11}};\\ Px_{i}=\sqrt{Q_{\left(2i-4)(2i-4\right)}};3\leq i\leq n\\ Py_{i}=\sqrt{Q_{\left(2i-3)(2i-3\right)}};3\leq i\leq n \end{array}$$

The error coefficient reflects the relationship between ranging error and positioning error. The smaller the error coefficient, the more accurate the positioning of the coordinates.

(ii)Root Mean Square Error

The Root Mean Square Error (RMSE) measures the deviation between the observed and the real values. The calculation formulas are as follows,
(41)$${\mathrm{RMSE}}_{x}=\frac{1}{N}\sqrt{\sum_{i=1}^{N}{\left(x_{real}-x_{i}\right)}^{2}}$$
(42)$${\mathrm{RMSE}}_{y}=\frac{1}{N}\sqrt{\sum_{i=1}^{N}{\left(y_{real}-y_{i}\right)}^{2}}$$
where *x_real_*, *y_real_* is the ideal coordinates, *x_i_*, *y_i_* is the measured coordinates. The deviation of the measured value of each x coordinates and y coordinates from the actual value can be obtained through the RMSE.

## 4. Simulation

The algorithm is used to calculate the base station’s coordinates by simulating the base station’s position and ranging error, and then the error coefficient and the RMSE of the coordinates is calculated.

According to the work [20,21], the shape and number of base stations will affect the positioning results. When the base station is set in a hexagonal shape, the positioning accuracy is relatively high. Therefore, in the simulation experiment, the base stations are arranged in a hexagonal shape. The coordinates of the six base stations are as follows, BS1 (0, 0, 2) BS2 (100, 0, 2), BS3 (25, 43.3, 2), BS4 (75, 43.3, 2), BS5 (−25, −43.3, 2), BS6 (−75, −43.3, 2).

Then, the influence of the selection of coordinate axis on the self-positioning accuracy of UWB movable base stations is analyzed; that is, BS1 is the origin of the coordinate system, and BS 1 and BS *n* are respectively the *x*-axis (*n* = 2, 3…6) (The coordinate axis is shown as the dotted part in Figure 5), analyze the accuracy of the measurement results of the base station position. Each base station performs 1000 positioning calculations and records the RMSE and error coefficient of the measured coordinates. Taking BS 5 as an example, the calculation result of the base station is shown in Figure 6. It can be seen that the error of the calculation result is ±0.04 m compared with the ideal coordinate, and the mean is within 0.01 m, the standard deviation is within 0.021 m, which meets the positioning needs.

It can be seen that the selection of the coordinate axis plays an considerable role in the positioning accuracy of base stations. Figure 7 and Figure 8 show that RMSE and error coefficient of the coordinate calculated under the coordinate system formed by taking BS1 as the origin and the line connecting BS 1 and BS *n* (*n* = 2, 3, 4, 5, 6…)as the *x*-axis. When BS 1 and 2 are used as the *x*-axis of the local coordinate system, the best measurement results are given. The maximum RMSE of the measurement results is 0.028 m, and the error coefficients are all less than 0.8. When BS 1 and 5 are used as the *x*-axis of the local coordinate system, the positional effect is poor. In this case, the coordinate error coefficients and the RMSE of the coordinate of BS 2 and BS 6 are larger. Because when two base stations are defined as the *x*-axis, the farther the distance is, the higher the degree of coincidence with the actual coordinate axis, the more accurate the positioning result will be.

Table 1 describes the ideal position and the simulation position of the base stations and the difference between them in the local coordinate system defining the *x*-axis by BS 1 and BS 2. It can be seen from the table that the y-coordinate error of BS 5 is the largest, which is 0.03 m. At the same time, the position error ($\sqrt{\Delta x^{2}+\Delta y^{2}}$) of the BS 5 is the largest, which is 0.032 m. For the base stations positioning of the sports field, the simulation result is within the accuracy requirement range, which verifies the feasibility of algorithm.

## 5. Experiments

In this section, we evaluate the positioning accuracy of the proposed algorithm through experiments on a 104 m × 70 m football field. First, the experimental setup is introduced, and then the experimental results of the algorithm in the sports field are shown. What’s more, the influence of different measurement methods on positioning accuracy is analyzed. Finally, we conducted an experiment on the positioning of people in the stadium. Our system consists of a hardware part and a software part. The hardware part includes 6 DecaWave DW1000 UWB transceivers, which can receive the ranging information between each other. Use STM32F405 microcomputer to control DW1000 for distance measurement through SPI communication. The software-controlled ranging process is as follows, firstly, BS 1 broadcasts the self-location command signal, and the other base stations perform ranging after receiving the command. The ranging result is sent to the computer by the BS 1; then, the BS1 in turn orders the remaining base stations broadcast positioning commands and receives the ranging information, and finally, the computer summarizes the distance information of each base station. As a result, the base stations communicate through the SDS-TWR mode, and only one base station’s signal is broadcast in each period time, which avoids interference between signals and confusion during the reception. In order to avoid the impact caused by personnel shielding, six UWB base stations are fixed in the middle of the upper goal frame of the football field, and all base stations operate under Line of Sight (LOS) conditions, as shown in Figure 9.

It can be seen from the simulation experiment in the previous section that the longer the distance between base stations, the more accurate the positioning results. Therefore, the experiment uses BS 1 and BS 2 to define the x-coordinate axis. After the base stations are fixed, use the total station to calibrate the coordinates of each base station (the positioning accuracy of the total stations is 1 mm + 2 ppm, and the distance measurement error of 1 km is 3 mm), through the translation and rotation of the coordinates, the calibration coordinates of the local coordinate system of the base station are obtained.

According to the work [22], the data transmission rate of the UWB signal and the communication distance are mutually restricted. The larger the data transmission rate, the closer the communication distance. The distance of 150 m on the sports field requires a lower transmission rate. For static positioning in LOS environment, reducing the transmission rate does not affect the positioning results. Therefore, the base stations of the system in this work use the 110 Kbps transmission rate for communication. It takes about 10 ms for the two base stations to perform a distance measurement. In the experiment, it only takes 0.2 s for the six base stations to complete the distance measurement. Compared with the manual deployment method, the time to obtain the position is shortened.

Table 2 and Table 3 show the self-positioning results of the base station in the football field experiment. Since the accuracy of the total station is 3 × 10^−6^ m, the calibration coordinates of the total station are taken as ideal coordinates. It can be seen from Table 2 that compared with the position calibrated by the total station, the maximum coordinate error of the coordinate measurement result is 0.05 m, which is 56.25% lower than the simulation result accuracy, the maximum coordinate error is 0.04 m, and the accuracy is within 0.05 m, the purpose of accurately positioning base stations in a large-scale environment of the sports field can be achieved. It can also be seen from Table 3 that the base station’s coordinate error coefficient is slightly higher than the simulated value. However both are less than 0.8, which can reduce the influence of the ranging error on the base stations positioning error. At the same time, the maximum value of RMSE is 0.041 m, which is 46.43% lower than the accuracy of the simulation result. However it is still within 0.05 m, which verifies the feasibility of the algorithm in this paper.

In order to further evaluate the tracking performance of the system, we conducted dynamic experiments on a 104 m × 70 m football field. The tester holds the positioning tag and walks along the calibrated rectangular track of the football field. It should be noted that because people hold tags while walking, there will inevitably be positioning results that are larger than the actual positioning error. We observe the effect of real-time positioning through the computer, and the positioning track is shown in Figure 10. And the position result of tag is shown in Figure 11, as can be seen from the figure, in the context of using the results of the fast-positioning algorithm as the base station coordinates for tag positioning, the average of the tag’s coordinate error is 0.28 m and the standard deviation is 0.15 m, which meets the positioning requirements of the sports field. It further explains the feasibility and practicability of the algorithm.

## 6. Conclusions

Since the positioning accuracy of athletes in the sports field is strictly related to the position accuracy of the base station, we designed a TOF-based fast self-positioning algorithm for UWB mobile base stations.

In order to improve the positioning accuracy, the coordinated positioning method of all mobile base stations is used to communicate with each other, which is solved by the Least Square method and the Newton Iteration method. Firstly, the error propagation law of the algorithm is derived in the established local coordinate system; Then, the method of reducing error is analyzed; finally, the feasibility of the algorithm is verified through simulation experiments and outdoor positioning experiments. The positioning results show that this method enables 6 base stations to perform rapid positioning within 0.2 s, with a positioning accuracy of 5 cm. Compared with manually measuring the position of UWB base stations, this algorithm does not require the use of high-precision instruments to assist measurement, which saves base station deployment time and solves the problem of the complex deployment process and high cost of traditional UWB mobile base stations.

## Figures and Tables

**Figure 1 sensors-21-06359-f001:**
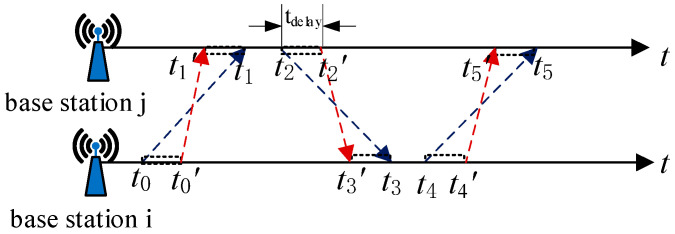
Principle diagram of SDS-TWR.

**Figure 2 sensors-21-06359-f002:**
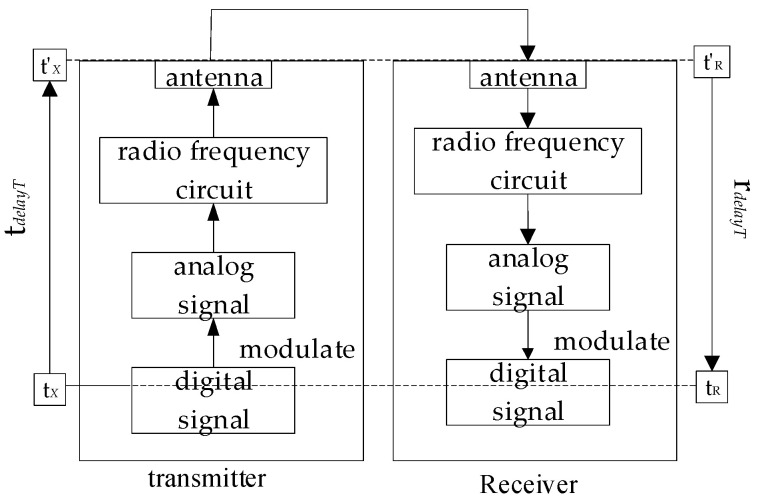
Decawave’s DWM1001 Development Board.

**Figure 3 sensors-21-06359-f003:**
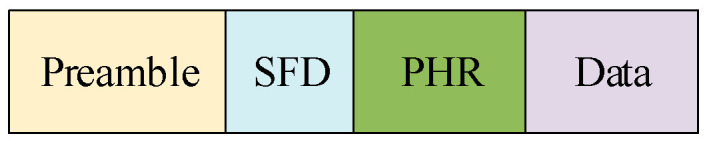
The frame structure of the physical layer.

**Figure 4 sensors-21-06359-f004:**
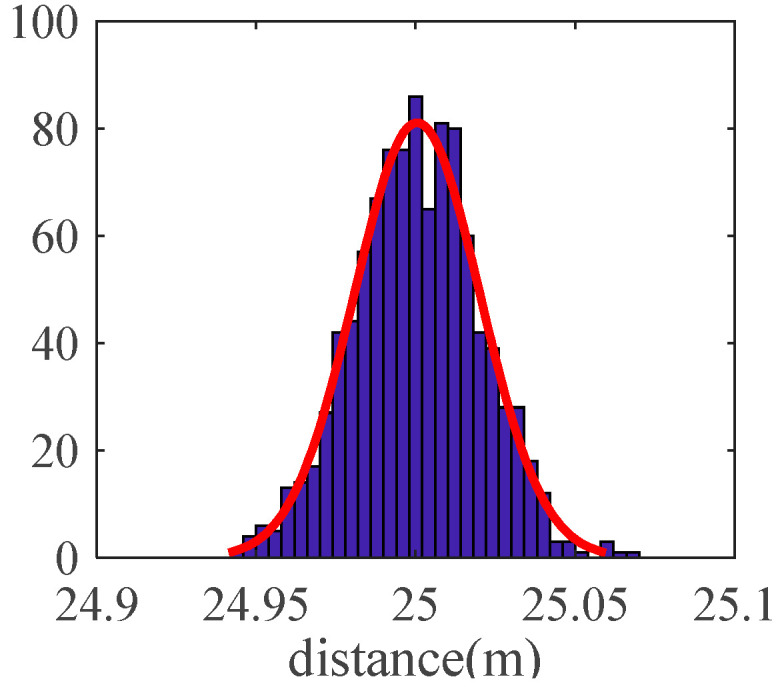
Ranging error obeys normal distribution.

**Figure 5 sensors-21-06359-f005:**
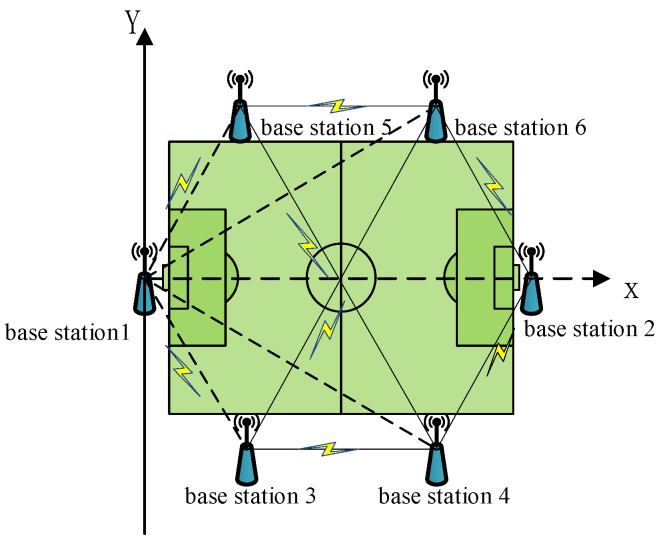
Schematic diagram of base stations rapid positioning.

**Figure 6 sensors-21-06359-f006:**
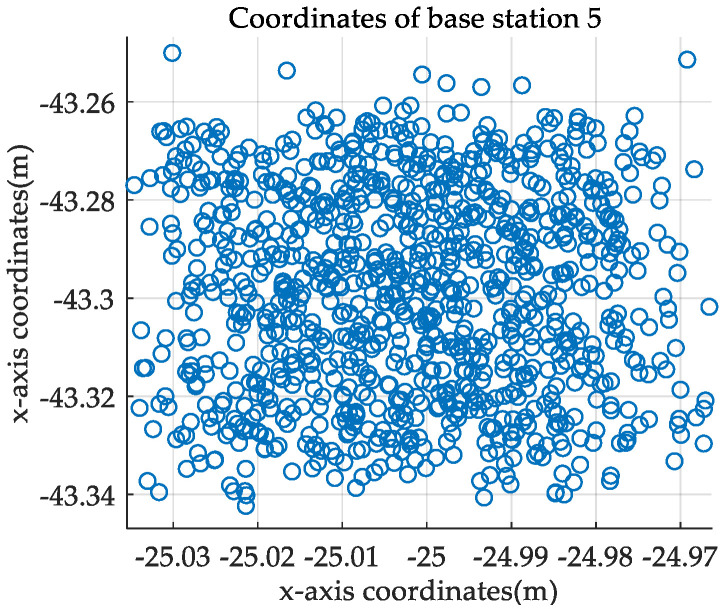
Schematic diagram of 1000 coordinate calculation results.

**Figure 7 sensors-21-06359-f007:**
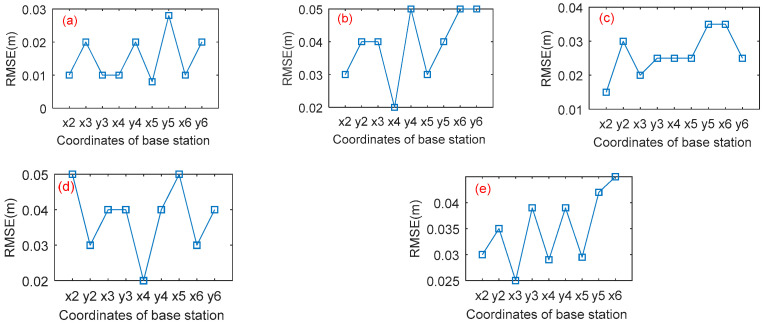
The RMSE of the hexagonal distribution ((**a**) the *x*-axis is determined by the BS 1 and 2; (**b**) the *x*-axis is determined by the BS 1 and 3; (**c**) the *x*-axis is determined by the BS 1 and 4; (**d**) the *x*-axis is determined by the BS 1 and 5; (**e**) the *x*-axis is determined by the BS 1 and 6).

**Figure 8 sensors-21-06359-f008:**
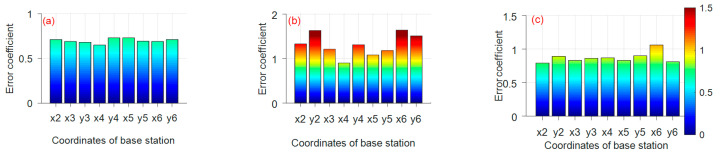
The error coefficient of the hexagonal distribution ((**a**) the *x*-axis is determined by the BS 1 and 2; (**b**) the *x*-axis is determined by the BS 1 and 3; (**c**) the *x*-axis is determined by the BS 1 and 4; (**d**) the *x*-axis is determined by the BS 1 and 5; (**e**) the *x*-axis is determined by the BS 1 and 6).

**Figure 9 sensors-21-06359-f009:**
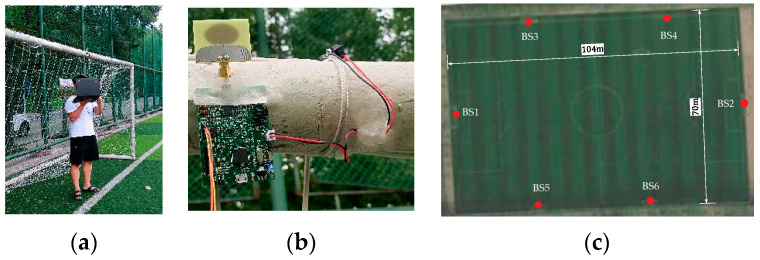
The environment of base station deployment ((**a**) receive the location information of the base station through the computer; (**b**) place the base station on the upper goal frame; (**c**) base station layout under satellite cloud image).

**Figure 10 sensors-21-06359-f010:**
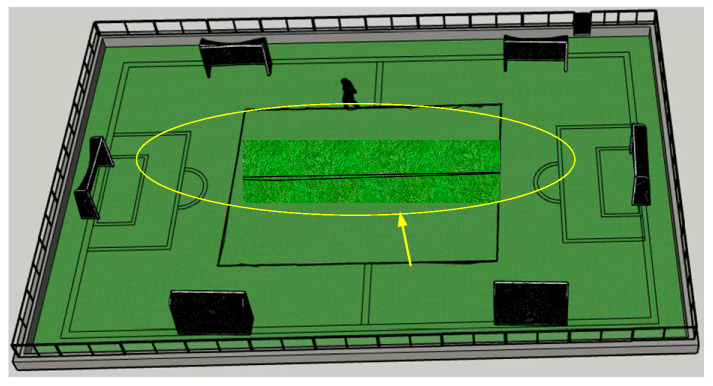
Movement trajectory of tester.

**Figure 11 sensors-21-06359-f011:**
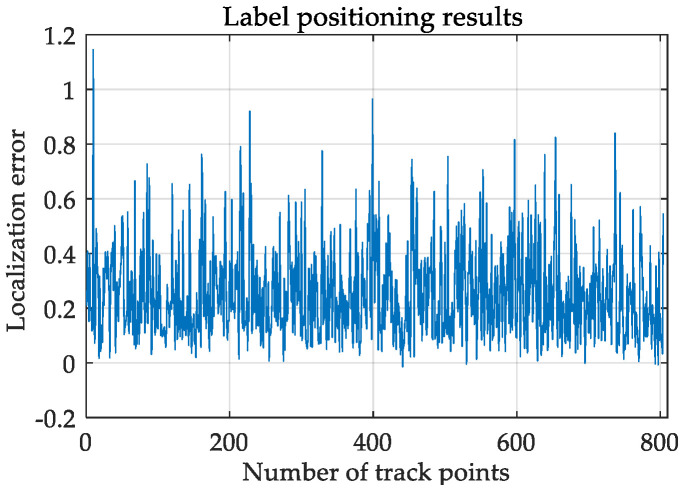
Label positioning results.

**Table 1 sensors-21-06359-t001:** Comparison of base station positioning coordinates and ideal position coordinates.

ID	Ideal Position (m)	Simulation Location (m)	Error (m)
2	(100, 0)	(99.99, 0)	(0.01, 0)
3	(25, 43.30)	(25.02, 42.29)	(0.02, 0.01)
4	(75, 43.30)	(75.01, 43.32)	(0.01, 0.02)
5	(−25, −43.30)	(−24.99, −43.33)	(0.01, 0.03)
6	(−75, −43.30)	(−75.01, −42.28)	(0.01, 0.02)

**Table 2 sensors-21-06359-t002:** Comparison of base station positioning coordinates and total stations positioning coordinates.

ID	Ideal Position (m)	Measurement Location (m)	Error (m)
2	(103.99, 0)	(103.96, 0)	(0.03, 0)
3	(27.18, 34.02)	(27.20, 34.05)	(0.02, 0.03)
4	(78.18, 35.11)	(78.20, 35.15)	(0.02, 0.04)
5	(27.19, −34.07)	(27.15, −34.10)	(0.04, 0.03)
6	(67.03, −34.34)	(67.07, −34.31)	(0.04, 0.03)

**Table 3 sensors-21-06359-t003:** Base stations coordinate error coefficient and root mean square error.

ID	Error Coefficient (m)	RMSE (m)
2	(0.70, 0)	(0.032, 0)
3	(0.68, 0.75)	(0.023, 0.034)
4	(0.72, 0.76)	(0.027, 0.041)
5	(0.75, 0.73)	(0.039, 0.035)
6	(0.76, 0.75)	(0.038, 0.025)

## Data Availability

All data, and codes related to the self positioning of the base station will be sent to the e-mail of the corresponding author upon request, and appropriate reasons will be provided.

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
