# Peer review of "TOF-Based Fast Self-Positioning Algorithm for UWB Mobile Base Stations"

_sensors, 2021, doi:10.3390/s21196359_

Round 1

Reviewer 1 Report

Line 39 is missing the definition for RFID, GNSS, RTK.... remember RTK is a sub class of GNSS. 

line 56 and 58 n should be italic. 

define DTDOA and TDOA inline 71 and 73.  

line 88 has one extra space before comma, please remove. 

Variables in section 2.1 (like i, t, j...) should be italic. 

line 154 SDS-TWR should be capital letters. 

line 187 Set the two base stations(BS) -> Set the two Base Stations(BS

what means Los in line 290?

in line 302 Firstly -> firstly 

line 323  Within ->  within

Section 3 and 4 should reorganized and better clarified, for example: test and materials and results discussion.  

Author Response

Dear Reviewer:

Thank you for your comments concerning our manuscript entitled “TOF-based Fast Self-positioning Algorithm for UWB Mobile Base Stations” (ID: 1351291). Those comments are all valuable and very helpful for revising and improving our paper, as well as the important guiding significance to our researches. We have studied comments carefully and have made correction which we hope meet with approval. Revised portion are marked in red in the paper. The main corrections in the paper and the responds to your comments are as flowing:

Point 1: Line 39 is missing the definition for RFID, GNSS, RTK.... remember RTK is a sub class of GNSS.

Response 1:

1)According to the reviewer’s suggestion, in order to improve the readability of the text, the acronym was changed to the full name. So we changed RFID into RFID (Radio Frequency Identification), and changed GNSS into GNSS (Global Navigation Satellite System).

2) Since RTK is a sub class of GNSS, we delete the term RTK and classify its references as GNSS.

Point 2: line 56 and 58 n should be italic.

Response2: We are very sorry for our negligence of the typeface, according to the reviewer’s suggestion, we have set n to italic.

Point 3: define DTDOA and TDOA inline 71 and 73. 

Response 3: According to the reviewer’s suggestion, in order to improve the readability of the text, the acronym was changed to the full name. So we changed DTDOA into Differential Time difference of arrival, and changed TDOA into Time difference of arrival.

Point 4: line 88 has one extra space before comma, please remove.

Response 4: We have made correction ac

cording to the Reviewer ’s comments.

Point 5: Variables in section 2.1 (like i, t, j...) should be italic.

Response 5: According to the reviewer’s suggestion, we have set the variable in italics.

Point 6: line 154 SDS-TWR should be capital letters.

Response 6: According to the reviewer’s suggestion, we have changed ‘sds-twr’ into ‘SDS-TWR’.

Point 7:line 187 Set the two base stations(BS) -> Set the two Base Stations(BS

Response 7: According to the reviewer’s suggestion, we have changed ‘base stations’ into ‘Base Stations’.

Point 8:what means Los in line 290?

Response 8: We have changed ‘LoS’ into ‘Line of Sight (LOS)’, and ‘LOS’ means there is no obstacle that affects the signal propagation between the two antennas, and the signal can be completely transmitted;

Point 9:in line 302 Firstly -> firstly

Response 9: We have changed ‘Firstly’ into ‘firstly’.

Point 10:line 323  Within ->  within

Response 10: We have changed ‘Within’ into ‘within’.

Point 11:Section 3 and 4 should reorganized and better clarified, for example: test and materials and results discussion. 

Response 11: In Section 3, we taking the coordinates of BS 5 as an example, we have added the coordinate distribution map of its 1000 position estimations, as in figure 6

In Section 4, we added a description of hardware and software, and added a label positioning experiment after base station positioning. And a small change was made to the conclusion.

Special thanks to you for your comments.

Reviewer 2 Report

The authors propose a Time of Flight (TOF) algorithm for the self-position of Ultra-Wideband mobile base stations. Applications of such systems are high-tech sports/smart sports which require a precise position to centimeter-level.

The proposed algorithm as opposed to the manual method with the use of tools is much more convenient, moreover when the accuracy of such measurement is within the accuracy of the manual measurement and comes with the advantage of rapid deployment.

The paper is well written and the theory holds. 

Minor revision to the following:

1. The correct TOF term mentioned in the Abstract is Time of Flight (TOF).

2. Acronyms for the abbreviated terms in the text could improve the readability of the text (eg Wifi, RFID, so on).

3. It would be great to test the algorithm in the future on moving targets(players). 

4. The experimental details could be improved and much better presented.
For example, the are only a few points of measurements for the experimental data, which are very close to the BS (base stations). Much more impact would be to consider other points than those considered in the paper or to be compared. The points can only be referenced to with detailed dimensions off the football field (length and width). After the description of the reference point (for example central football field as (0,0)) for cartesian representation, the points of experimental measurement can be precisely reported (x, y).

5. Also, for comparison, it has to be expressed what precision has the tool to describe the ideal location mentioned in table 2. In this way, the reader would have the sense of comparison with the distance obtained with the algorithm as reported in the measurement location column in table 2 and the associated error.

6. The is also the mention that 1000 position calculations are done before reporting the distance estimation for each base station. It would greatly improve the paper if a distribution of the measured errors (1000 positions) would be represented. 

Good luck with your research.

Author Response

List of Responses

Dear Reviewer:

Thank you for your comments concerning our manuscript entitled “TOF-based Fast Self-positioning Algorithm for UWB Mobile Base Stations” (ID: 1351291). Those comments are all valuable and very helpful for revising and improving our paper, as well as the important guiding significance to our researches. We have studied comments carefully and have made corrections which we hope meet with approval. Revised portions are marked in red on the paper. The main corrections in the paper and the responses to your comments are as flowing:

Point 1: The correct TOF term mentioned in the Abstract is Time of Flight (TOF).

Response 1: We are very sorry for our incorrect writing, we have changed ‘Time of Fly’ into ‘Time of Flight’.

Point 2: Acronyms for the abbreviated terms in the text could improve the readability of the text (eg Wifi, RFID, so on).

Response 2: We have made corrections according to the reviewer’s comments. In order to improve the readability of the text, the acronym was changed to the full name. So we changed RFID into RFID (Radio Frequency Identification), and changed GNSS into GNSS (Global Navigation Satellite System), and changed Wifi into Wifi (Wireless Fidelity)

Point 3:  It would be great to test the algorithm in the future on moving targets(players).

Response 3: we have added experiments on the positioning of people in the stadium,

Point 4:  The experimental details could be improved and much better presented.

For example, the are only a few points of measurements for the experimental data, which are very close to the BS (base stations). Much more impact would be to consider other points than those considered in the paper or to be compared. The points can only be referenced to with detailed dimensions off the football field (length and width). After the description of the reference point (for example central football field as (0,0)) for cartesian representation, the points of experimental measurement can be precisely reported (x, y).

Response 4: Thank you very much for your suggestions,  my responses are as follows:

1) Since the algorithm is aimed at the location of the base station, the positioning result is around the base station, indicating the accuracy of the positioning result. Considering the reviewer's suggestion, we have added the test of the tester's motion track on the playground to further illustrate the feasibility of the algorithm.

2) Considering the actual situation of the sports field and the practicality of the system, placing the base station on the side of the sports field is the best choice, and placing it inside the sports field will affect the athletes.

3)We think to take central football field as (0,0) is very interested in this proposal, but the focus of this paper is to establish a local coordinate system in the algorithm, we want to keep the same focus, but we have your point of view as future solution method of applications.

Point 5:  Also, for comparison, it has to be expressed what precision has the tool to describe the ideal location mentioned in table 2. In this way, the reader would have the sense of comparison with the distance obtained with the algorithm as reported in the measurement location column in table 2 and the associated error.

Response 5: As mentioned in line 321, (the positioning accuracy of the total stations is: 1mm +2 ppm, and the distance measurement error of 1km is 3mm), and we add the following sentence before Table 2, Since the accuracy of the total station is 3×10-6 m, the calibration coordinates of the total station are taken as ideal coordinates.

Point6:The is also the mention that 1000 position calculations are done before reporting the distance estimation for each base station. It would greatly improve the paper if a distribution of the measured errors (1000 positions) would be represented.

Response 6: According to the reviewer’s suggestion, taking the coordinates of the base station 5 as an example, we have added the coordinate distribution map of its 1000 position estimations, as in figure 6

Special thanks to you for your comments.

Round 2

Reviewer 1 Report

The document was significantly improved and more clear.  

Author Response

Dear reviewer,

First of all, thank you very much for your recognition of our paper. Then, thank you again for your suggestions for improving our paper!

Reviewer 2 Report

Dear Authors,

The manuscript in the new version is much improved. But, there are still some missing data from the description of experimental measurements as the length and width of the football stadium. 

The reader may guess that is about 105 meters (length) x 68 meters  (width). However, it would be great if you can provide this information.

Figure 10, 11, the experimental movement trajectory testing greatly improves your results. 

Good luck on your further research!

Author Response

List of Responses

Dear Reviewer:

Thank you for your comments concerning our manuscript entitled “TOF-based Fast Self-positioning Algorithm for UWB Mobile Base Stations” (ID: 1351291). We have studied comments carefully and have made corrections which we hope meet with approval. The main corrections in the paper and the responses to your comments are as flowing:

Point 1: The manuscript in the new version is much improved. But, there are still some missing data from the description of experimental measurements as the length and width of the football stadium.

Response 1: We added a description of the length and width of the football field in the introduction to the experiment in line 296.

Point 2: The results and the conclusions can be improved

Response 2:

1. We have added the specific mean and standard deviation in 1000 position measurements in line 260.

2. We have improved the conclusions. First, we streamlined the conclusions. Secondly, we added the accuracy and rapidity reflected in the experimental results to the conclusions.

Special thanks to you for your comments.